

# Climatology of the mesopause density using a global distribution of meteor radars

Wen Yi[1,2], Xianghui Xue[1,2,5], Iain M. Reid[3,4], Damian J. Murphy[6], Chris M. Hall[7],

Masaki Tsutsumi[8], Baiqi Ning[9], Guozhu Li[9], Robert A. Vincent[3,4], Jinsong Chen[10],

Jianfei Wu[1,2], Tingdi Chen[1,2], Xiankang Dou[1]

[1]CAS Key Laboratory of Geospace Environment, Department of Geophysics and
Planetary Sciences, University of Science and Technology of China, Hefei, China.

[2]Mengcheng National Geophysical Observatory, School of Earth and Space Sciences,
University of Science and Technology of China, Hefei, China

[3]ATRAD Pty Ltd., Thebarton, South Australia, Australia,

[4]School of Physical Sciences, University of Adelaide, Adelaide, South Australia,
Australia

[5]Synergetic Innovation Center of Quantum Information and Quantum Physics,
University of Science and Technology of China, Hefei, China

[6]Australian Antarctic Division, Kingston, Tasmania, Australia

[7]Tromsø Geophysical Observatory, UiT – The Arctic University of Norway, Tromsø,
Norway

[8]National Institute of Polar Research, Tachikawa, Japan

[9]Key Laboratory of Earth and Planetary Physics, Institute of Geology and Geophysics,
Chinese Academy of Sciences, Beijing, China

[10]National Key Laboratory of Electromagnetic Environment, China Research Institute
of Radiowave Propagation, Qingdao, China

Corresponding author: Xianghui Xue and Iain M. Reid (xuexh@ustc.edu.cn,
ireid@atrad.com.au)





**Abstract:** The existing distribution of meteor radars located from high- to low-latitude regions provides a favourable temporal and spatial coverage for investigating the climatology of the global mesopause density. In this study, we report the climatology of the mesopause density estimated using multiyear observations from nine meteor radars, namely, the Davis Station (68.6 °S, 77.9 °E), Svalbard (78.3 °N, 16 °E) and Tromsø (69.6 °N, 19.2 °E) meteor radars located at high latitudes, the Mohe (53.5 °N, 122.3 °E), Beijing (40.3 °N, 116.2 °E), Mengcheng (33.4 °N, 116.6 °E) and Wuhan (30.5 °N, 114.6 °E) meteor radars located in the mid-latitudes, and the Kunming (25.6 °N, 103.8 °E) and Darwin (12.3 °S, 130.8 °E) meteor radars located at low latitudes. The daily mean density was estimated using ambipolar diffusion coefficients derived from the meteor radars and temperatures from the Microwave Limb Sounder (MLS) on board the Aura satellite. The seasonal variations in the Davis Station meteor radar densities in the southern polar mesopause are mainly dominated by an annual oscillation (AO). The mesopause densities observed by the Svalbard and Tromsø meteor radars at high latitudes and the Mohe and Beijing meteor radars at high mid-latitudes in the Northern Hemisphere show mainly an AO and a relatively weak semiannual oscillation (SAO). The mesopause densities observed by the Mengcheng and Wuhan meteor radars at lower mid-latitudes and the Kunming and Darwin meteor radars at low latitudes show mainly an AO. The SAO is evident in the Northern Hemisphere, especially at high latitudes, and its largest amplitude, which is detected at the Tromsø meteor radar, is comparable to the AO amplitudes. These observations indicate that the mesopause densities over the southern and northern high latitudes exhibit a clear seasonal asymmetry. The maxima of the yearly variations in the mesopause densities display a clear temporal variation across the spring equinox as the latitude decreases; these latitudinal variation characteristics may be related to latitudinal changes influenced by gravity wave forcing. In addition to an AO, the mesopause densities over low latitudes also clearly show a variation with a periodicity of 30-60 days related to the Madden-Julian oscillation in the subtropical troposphere.




## 1. Introduction

The temperatures, winds and densities in the mesopause region are essential for studying the dynamics and climate, including both short-term wave motions (e.g., gravity waves, tides, and planetary waves) and long-term climate variations (e.g.,

interannual variations, seasonal variations and intraseasonal variations), of the middle and upper atmosphere. The climatology of the temperature and wind within the mesopause region has been studied for decades using ground-based instruments such as meteor radars, medium-frequency (MF) radars, lidars (Dowdy et al., 2001; Dou et al., 2009; Li et al., 2008; 2012; 2018; Li et al., 2015) and satellite instruments (Garcia

et al., 1997; Remsberg et al., 2002; Xu et al., 2007). It is well established that the semiannual oscillation (SAO) dominates the seasonal variations in both the wind and the temperature in the low-latitude mesosphere (Li et al., 2012), whereas the annual oscillation (AO) dominates the seasonal variations in the mid- and high-latitude mesosphere (Remsberg et al., 2002; Xu et al., 2007; Dou et al., 2009). However, in

contrast to temperature and wind observations, long-term continuous measurements of the atmospheric density in the mesopause region are still quite rare; as a result, the seasonal variations in the mesopause, especially with regard to its global structure, are still unclear.

Meteor radar operates both day and night under all kinds of weather and geographical

conditions and provides good long-term observations; consequently, meteor radar is a powerful technique for studying the dynamics and climate of the mesopause region, including its wind fields and temperatures (e.g., Hocking et al., 2004; Holdsworth et al., 2006; Hall et al., 2006, 2012; Stober et al., 2008, 2012; Yi et al., 2016; Lee et al., 2016; Holmen et al., 2016; Liu et al., 2017; Lima et al., 2018; Ma et al., 2018). In

addition to acquiring wind and temperature measurements, meteor radar has also been applied in recent years to estimate the atmospheric density in the mesopause region. For instance, the variation in the peak height of meteor radar detections can be used to estimate changes in the mesopause density (e.g., Clemesha and Batista, 2006; Stober et al., 2012, 2014; Lima et al., 2015; Liu et al., 2016). However, the seasonal



variations in the peak height are not affected by the atmospheric density alone; they are also significantly influenced by the properties of meteoroids, especially the meteor velocity (Stober et al., 2012; Yi et al., 2018b). Furthermore, the mesospheric densities can also be estimated from meteor radar-derived ambipolar diffusion coefficients, and

the mesospheric temperatures can be derived from other measurements (e.g., Takahashi et al., 2002; Yi et al., 2018b). Therefore, in this study, we apply ambipolar diffusion coefficients derived from a global distribution of meteor radars in addition to temperature measurements simultaneously obtained by the Microwave Limb Sounder (MLS) on board the Aura satellite to determine the mesopause density. In addition,

long-term observations of global atmospheric densities are used to study the latitudinal and seasonal variations in the mesopause region. Descriptions of the instrument datasets, the method, and the error estimation approach are presented in section 2. Then, the seasonal variations in the mesopause density are presented in section 3, followed by a composite analysis in section 4. Finally, a summary is

provided in section 5.

## 2. Data and methods

In this study, data from nine meteor radars, namely, the Davis Station (68.6 °S, 77.9 °E), Svalbard (78.3 °N, 16 °E), Tromsø (69.6 °N, 19.2 °E), Mohe (53.5 °N, 122.3 °E), Beijing (40.3 °N, 116.2 °E), Mengcheng (33.4 °N, 116.5 °E), Wuhan (30.6 °N, 114.4 °E),

Kunming (25.6 °N, 108.3 °E), and Darwin (12.3 °S, 130.5 °E) meteor radars (hereinafter referred to as DMR, SMR, TMR, MMR, BMR, McMR, WMR, KMR and DwMR, respectively), were used. Table 1 summarizes the operational frequencies, geographic locations and observational time periods for the meteor radars used in this study. These meteor radars all belong to the ATRAD meteor detection radar (MDR) series

and are similar to the Buckland Park meteor radar system described by Holdsworth et al. (2004). Figure 1 shows the locations of these nine meteor radars. The SMR and TMR are located in the northern high latitudes, whereas the MMR, BMR, McMR and WMR are positioned in the northern mid-latitudes, and the KMR is situated in the northern low latitudes. In contrast, we have only two meteor radars, namely, the DMR





located in the southern high latitudes and the DwMR situated in the southern low latitudes, in the Southern Hemisphere because it is covered primarily by oceans.

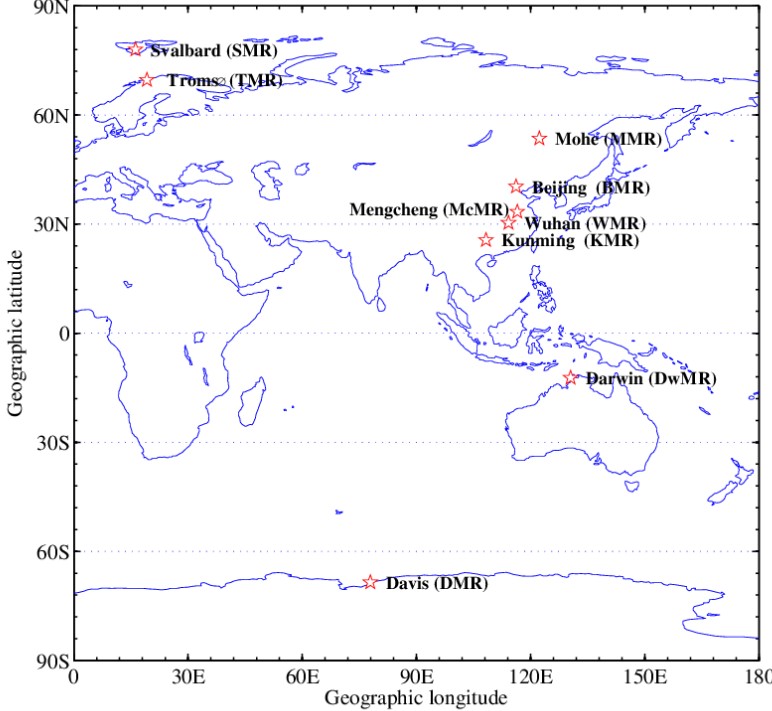

**Figure 1.** The locations of the meteor radars used in this study.

The ambipolar diffusion coefficient ($D_a$) observed by a meteor radar describes the rate at which plasma diffuses in a neutral background and is a function of both the atmospheric temperature, $T$, and the atmospheric density, $\rho$, as given by

$$\rho = 2.23 \times 10^{-4} K_0 \frac{T}{D_a}, \tag{1}$$

where $K_0$ is the ionic zero-field mobility, which is assumed to be $2.5 \times 10^{-4}\ m^{-2}s^{-1}V^{-1}$ (Hocking et al., 1997). Using the relation given by equation (1), measurements of the temperature and $D_a$ from meteor radars can be used to retrieve the neutral mesospheric density (see, e.g., Takahashi et al., 2002; Yi et al., 2018b).



**Table 1.** Main operation parameters, geographic coordinates and observational time periods for the meteor radars used in this study.

| | Meteor radar | Geographic coordinates | Frequency | Data used in this study |
|---|---|---|---|---|
| Northern hemisphere | Svalbard (SMR) | 78.3 °N, 16 °E | 31 MHz | 2005.01-2016.12 |
| | Tromsø (TMR) | 69.6 °N, 19.2 °E | 30.3 MHz | 2005.01-2016.12 |
| | Mohe (MMR) | 53.5 °N, 122.3 °E | 38.9 MHz | 2011.08-2018.04 |
| | Beijing (BMR) | 40.3 °N, 116.2 °E | 38.9 MHz | 2011.01-2018.04 |
| | Mengcheng (McMR) | 33.4 °N, 116.5 °E | 38.9 MHz | 2014.09-2018.04 |
| | Wuhan (WMR) | 30.6 °N, 114.4 °E | 38.9 MHz | 2012.10-2017.08 |
| | Kunming (KMR) | 25.6 °N, 108.3 °E | 37.5 MHz | 2011.04-2014.12 |
| Southern hemisphere | Davis (DMR) | 68.6 °S, 77.9 °E | 33.2 MHz | 2005.01-2016.12 |
| | Darwin (DwMR) | 12.3 °S, 130.5 °E | 33.2 MHz | 2006.01-2009.12 |

The MLS instrument on board the Earth Observing System (EOS) Aura spacecraft was launched in 2004. For this investigation, the Aura MLS temperature (Schwartz et al., 2008) and geopotential height data (version 4) were restricted to data obtained within a $10° \times 20°$ bounding box centred on each of the abovementioned meteor radar locations. The daily averaged MLS temperature and geopotential height observations were interpolated into 1 km bins between 85 and 95 km to produce temperature profiles using geometric heights obtained from geopotential heights (Yi et al., 2018b).

In this study, the daily neutral mesospheric densities from 85 to 95 km were estimated using the daily mean $D_a$ from the nine meteor radars and the Aura MLS temperatures using equation (1). Yi et al. (2018b) showed that the $\log_{10} D_a$ profiles derived from meteor radars are approximately linear with respect to the altitude in the range from 85 to 95 km, which indicates that mainly ambipolar diffusion governs the evolution of meteor trails in this region. In general, the $\log_{10} D_a$ profiles measured by meteor radars have larger slopes than those derived from Sounding of the Atmosphere using



Broadband Emission Radiometry (SABER) (Yi et al., 2018b) and MLS (Younger et al., 2015) measurements. To avoid the influence of the bias in $D_a$, in the present study, we use the relative variation in the density to examine the climatology of the global mesopause density.

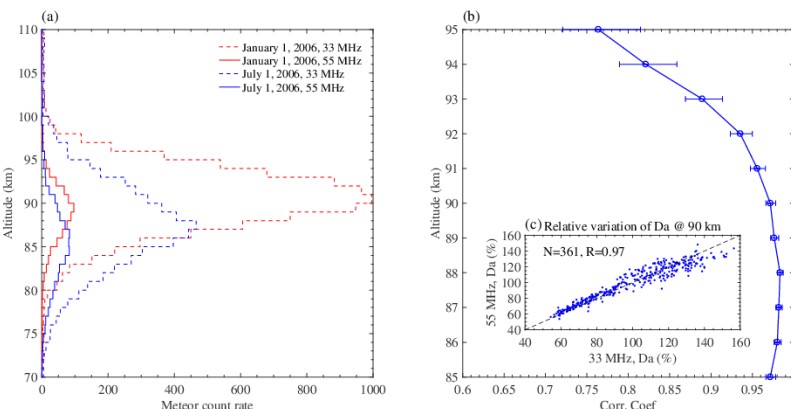

**Figure 2.** (a) The height distributions of meteor detections in 1 km bins on January 1 and July 1 in 2006 from the 33 MHz and 55 MHz meteor radars at Davis Station. (b) Height variation in the correlation coefficient between the $D_a$ observed simultaneously by the 33 MHz and 55 MHz meteor radars at Davis Station in 2006.

The error bars indicate the lower and upper bounds of the 95% confidence interval for each coefficient. (c) Comparison of the variations in the daily mean $D_a$ (blue dots) at 90 km simultaneously observed from the 33 MHz and 55 MHz meteor radars at Davis Station in 2006. The percentage variations in $D_a$ with respect to the yearly mean $D_a$ in 2006. N represents the number of days observed by these two radars in 2006, and R

denotes the linear correlation coefficient.

There is an uncertainty in $D_a$ caused by the estimation of the decay time of meteor echoes (e.g., Cervera and Reid, 2000; Holdsworth et al., 2004); unfortunately, this uncertainty is quite difficult to estimate from the radar system directly. In addition, the number of precise, simultaneously observed temperature and density measurements in

the study region is insufficient to estimate the absolute error in $D_a$ through a comparison. Yi et al. (2018b) compared simultaneous observations of $D_a$ acquired by



two co-located meteor radars at Kunming and found that the relative errors in the daily mean $D_a$ and the density at 90 km obtained from the KMR should be less than 5% and 6%, respectively. Here, to estimate the relative errors in $D_a$ and the density, we conduct a similar approach using simultaneous meteor echoes observed by two

co-located meteor radars with different frequencies (33 MHz and 55 MHz) at Davis Station. The 33 MHz and 55 MHz meteor radars at Davis Station are described in related studies (see, e.g., Reid et al., 2006; Younger et al., 2014).

Figure 2a shows the height distributions of meteor detections in 1 km bins on January 1 and July 1 in 2006 for the 33 MHz and 55 MHz meteor radars at Davis Station. The

meteor count observed by the 55 MHz meteor radar is much lower than that observed by the 33 MHz meteor radar because the former is a mesosphere-stratosphere-troposphere (MST) radar operating with time-interleaved stratosphere-troposphere (ST), meteor, and polar mesosphere summer echoes (PMSE) modes (see, e.g., Reid et al., 2006). Figure 2b shows the correlation coefficients

between the $D_a$ observed simultaneously by the two co-located meteor radars from 85 to 95 km. The correlation coefficients are higher than 0.96 below 92 km, and they become lower as the altitude increases above 92 km; this occurs mainly because the meteor count (as shown in Figure 2a) obtained by the 55 MHz meteor radar above 92 km is too low to provide a good precision in $D_a$. The strong correlation between the

$D_a$ measurements from the two independent meteor radars indicates that the variations in $D_a$ are dominated by the same geophysical variations (i.e., gravity waves, tides and planetary waves) from below as well as by disturbances by geomagnetic forcing from above (Yi et al., 2017, 2018a) rather than by random systemic errors; therefore, the difference between the two $D_a$ measurements is considered to be representative of the

relative uncertainty in $D_a$.

**Table 2. The relative uncertainties in $D_a$ and the density from 85 to 95 km**

| Altitude (km) | 85 | 86 | 87 | 88 | 89 | 90 | 91 | 92 | 93 | 94 | 95 |
|---|---|---|---|---|---|---|---|---|---|---|---|
| Relative uncertainties in $D_a$ (%) | 3 | 2 | 2 | 2 | 2 | 3 | 4 | 6 | 11 | 18 | 24 |
| Relative uncertainties in the density (%) | 3.6 | 2.8 | 2.8 | 2.8 | 2.8 | 3.6 | 4.5 | 6.3 | 11.2 | 18.1 | 24 |





The MLS temperature has an accuracy of 1-3 K from 316 hPa to 0.001 hPa (Schwartz et al., 2008); thus, the uncertainty in the density induced by the MLS temperature uncertainty would be less than 2% based on the present values. Table 2 shows a summary of the relative uncertainties in the density from 85 to 95 km. The density uncertainties are less than 6% below 92 km and become larger as the altitude increases above 92 km. However, under real-world conditions, the meteor counts from the nine meteor radars used in this study are much larger than those from the 55 MHz meteor radar, and hence, it is reasonable to believe that the uncertainties in the density above 92 km would be lower than this estimate of 6%.

**3. Seasonal variations in the global mesopause density**

Figure 3 shows the monthly mean mesopause densities in the southern polar region derived from the DMR and in the northern polar region derived from the SMR and TMR between 2005 and 2016. As shown in Figure 3a, the DMR densities are dominated by an AO with a maximum during the spring and a minimum during the early winter. The annual variations in the DMR densities are approximately 65% of the mean density. Younger et al. (2015) developed a novel technique using meteor radar echo decay times from the DMR to determine the height of a constant-density surface in the mesopause region and found that the height of the constant-density surface is also dominated by an AO. In the northern polar region, the SMR densities mainly show an AO and a relatively weak SAO with a clear maximum during the spring. However, the minima of the SMR densities are not as regular as the DMR densities, and they appear approximately during the summer and winter. The TMR densities mainly show an AO and SAO with a clear maximum during the spring and two distinct minima during the summer and winter. As the SMR and TMR are in the northern polar region, the SMR and TMR densities show a similar annual variation; however, the semiannual variations in the TMR densities are more obvious than those of the SMR densities.

To further examine the periodicities present in the mesopause densities derived from the meteor radars, Lomb-Scargle periodograms were calculated for the entire



observational period of the densities in each 1 km bin from 85 to 95 km. Figure 4 shows the contours of the Lomb-Scargle periodograms of the mesopause densities obtained from the DMR, SMR and TMR. The periodograms of the DMR densities in Figure 4a are clearly dominated by an AO as well as a relatively weak SAO; the

largest amplitude of the AO appears at 87 km, where it is 20% of the mean DMR densities. The periodograms of the SMR densities mainly show an AO and SAO, and the amplitudes of the AO and SAO at 90 km are approximately 12% and 8%, respectively, of the mean SMR densities. The TMR mainly shows an AO and SAO, and the amplitudes of the AO and SAO at 90 km are approximately 11% and 10%,

respectively. In addition to an AO and SAO, the northern polar mesospheric densities from the SMR and TMR also exhibit clear seasonal periodicities with quasi-120-day and quasi-90-day oscillations.

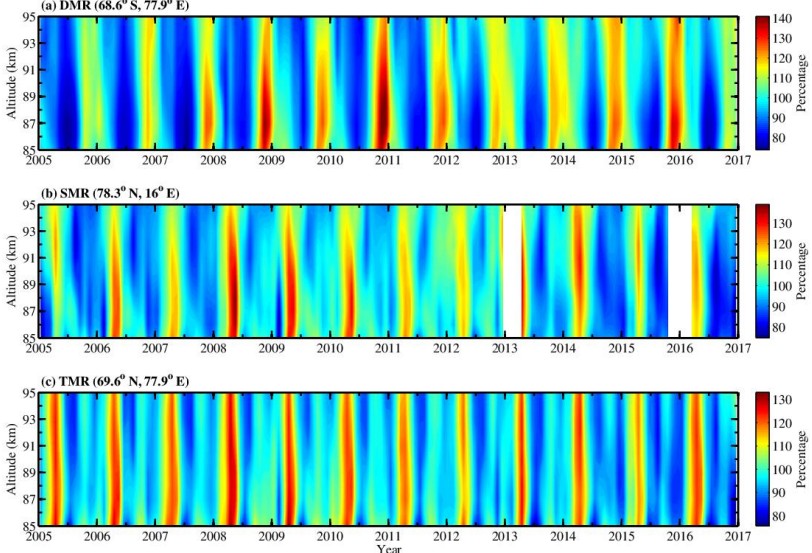

**Figure 3.** Variations in the monthly mean densities at altitudes from 85 to 95 km

obtained from the DMR, SMR, and TMR between 2005 and 2016. The colour bars indicate the percentage variation in the monthly mean density relative to the mean density from the total observational time period.





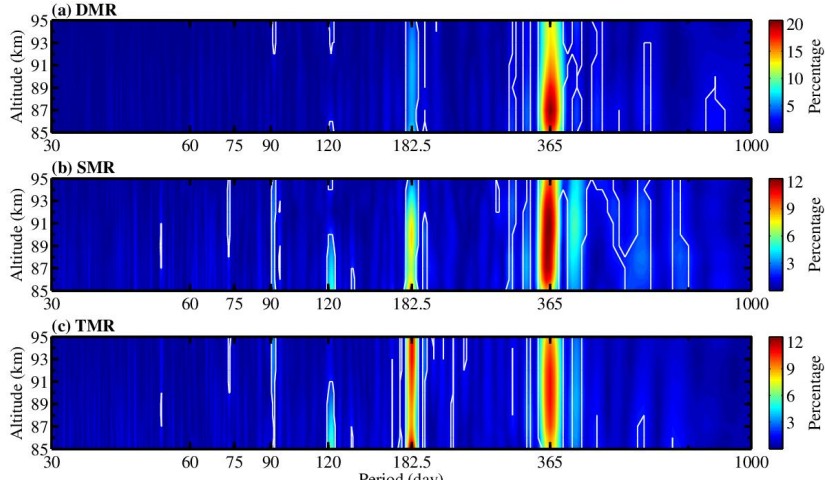

**Figure 4.** Contours of the Lomb-Scargle spectral (see, e.g., Lomb, 1976; Scargle, 1982) amplitudes of the (a) DMR, (b) SMR and (c) TMR densities. The white lines represent the 99% significance level.





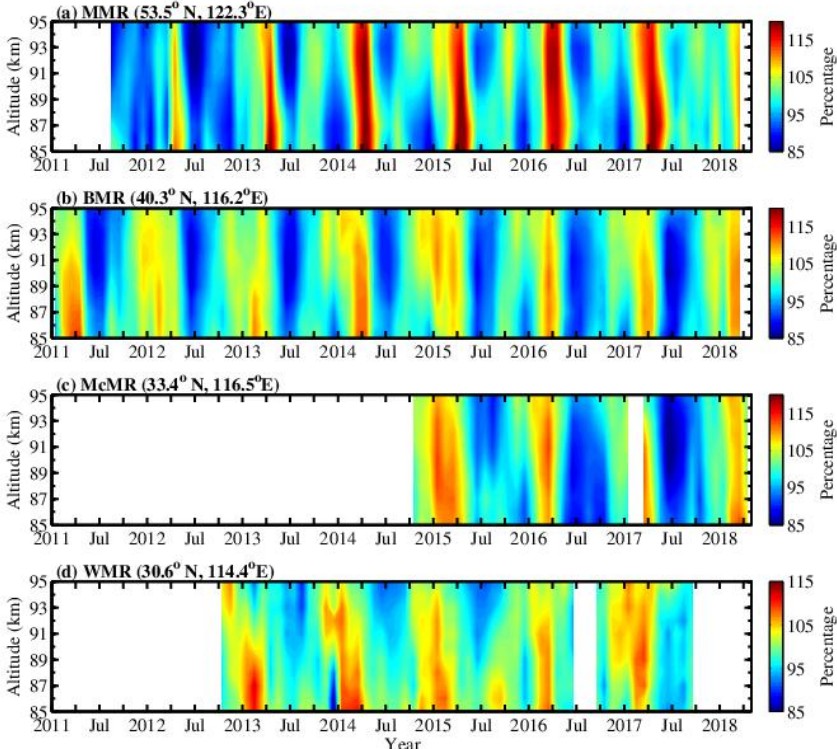

**Figure 5.** Same as Figure 3, but for the MMR, BMR, McMR and WMR monthly mean densities.

Figure 5 shows the monthly mean mesospheric densities at northern mid-latitudes derived from the MMR, BMR, McMR and WMR. The MMR monthly mean densities (Figure 5a) from August 2012 to April 2018 at higher mid-latitudes clearly show both an AO and an SAO; the AO clearly reaches a maximum in the spring (April), whereas the SAO shows two distinct minima: one clearly appears in the summer above 90 km, and another clearly appears in the winter below 90 km. As shown in Figure 5b, the BMR monthly mean densities from January 2011 to April 2018 show mainly an AO with a maximum during the spring and a minimum during the summer. The McMR monthly mean densities (Figure 5c) from October 2014 to April 2018 show seasonal variations similar to those exhibited by the BMR densities with a clear minimum



during the summer and a maximum during the spring. As shown in Figure 5d, the WMR monthly mean densities from October 2012 to September 2017 show mainly an AO with a maximum during the late winter and a minimum during the summer. As the WMR is located close to the low latitudes, the annual variations in the WMR densities

are much smaller than those in the densities observed by meteor radars at high latitudes and higher mid-latitudes.

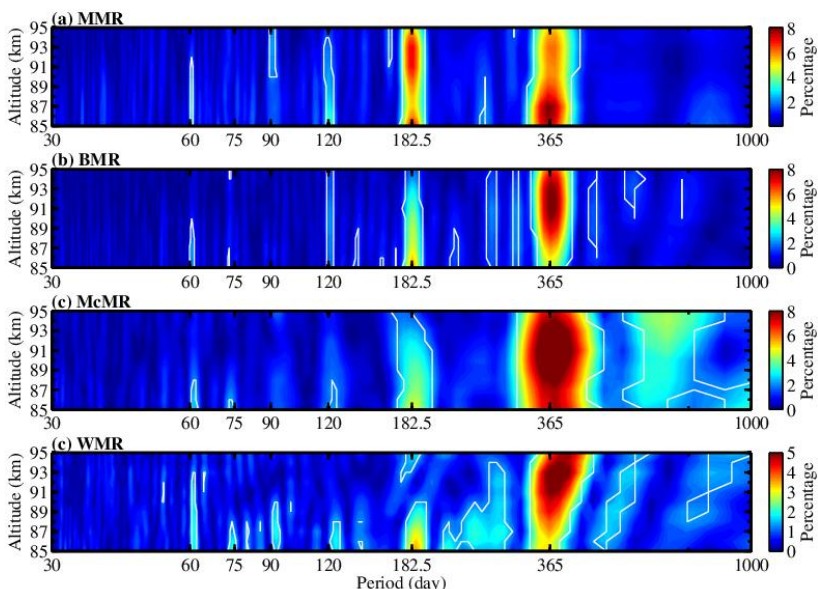

**Figure 6.** Same as Figure 4, but for the MMR, BMR, HMR and WMR daily mean densities in the mid-latitudes.

Figure 6 displays the contours of the Lomb-Scargle periodograms of the mesopause densities from the MMR, BMR, McMR and WMR. The MMR densities (Figure 6a) show mainly an AO and SAO; the amplitudes of the AO reach a maximum at 87 km, where the amplitude is approximately 8% of the MMR mean densities, while the amplitudes of the SAO are larger than those of the AO above 90 km with a maximum

that is approximately 7% of the MMR mean densities at 93 km. The BMR and McMR densities (Figure 6b and 6c, respectively) show similar periodograms; they exhibit mainly an AO and a relatively weak SAO. In contrast, the WMR densities are



dominated by an AO above 89 km; however, below 89 km, they show both an SAO
and an AO.

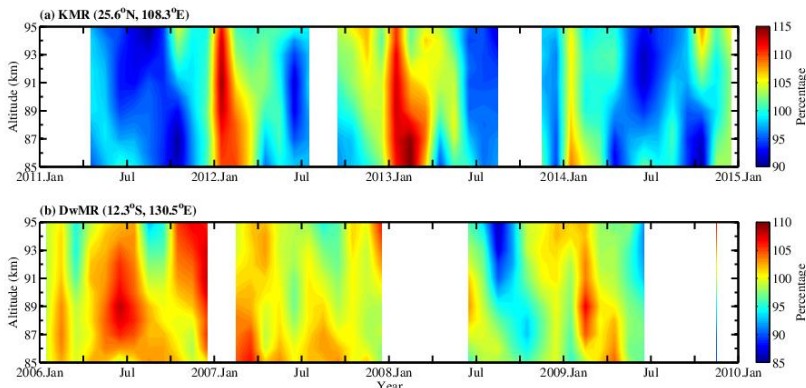

**Figure 7.** Same as Figure 3, but for the KMR and DwMR monthly mean densities at
low latitudes.

Figure 7a shows the KMR densities in the northern low latitudes from April 2011 to
December 2014. The KMR densities show mainly an AO with a maximum during the
winter and a minimum during the summer. Figure 7b shows the DwMR densities at
southern low latitudes from January 2006 to June 2009. The DwMR densities exhibit
a large data gap; however, the data still provide the opportunity to investigate the
climatology of the mesospheric density at southern low latitudes. The seasonal
variations in the DwMR densities are more complicated than those in the KMR
densities and clearly show intraseasonal (with a periodicity of 30-60 days) oscillations.
To more clearly examine the seasonal variations in the mesospheric densities, Figure 8
shows the Lomb-Scargle periodograms of the KMR and DwMR densities. The largest
component of the KMR densities is an AO above 87 km, followed by an SAO, a
90-day oscillation and a 60-day oscillation; below 87 km, the SAO becomes more
obvious in the KMR densities, which can also be seen in Figure 7a. The DwMR
densities show both an AO and an SAO above 92 km. In addition to seasonal
variations, the DwMR densities also exhibit broad oscillations with periodicities
ranging from 30 to 60 days; these periodic variations may be similar to intraseasonal




oscillations (Eckermann and Vincent, 1994), which are related to the Madden-Julian oscillation (MJO) in the tropical troposphere (please refer to Madden and Julian, 1974). We also examined the zonal mean winds observed by the DwMR and found an oscillation with a similar periodicity (30-60 days) in the zonal mean winds within the mesopause. This observation reveals that the lower and middle atmosphere is coupled; however, a more detailed discussion of this intercomparison will be described in a forthcoming paper.

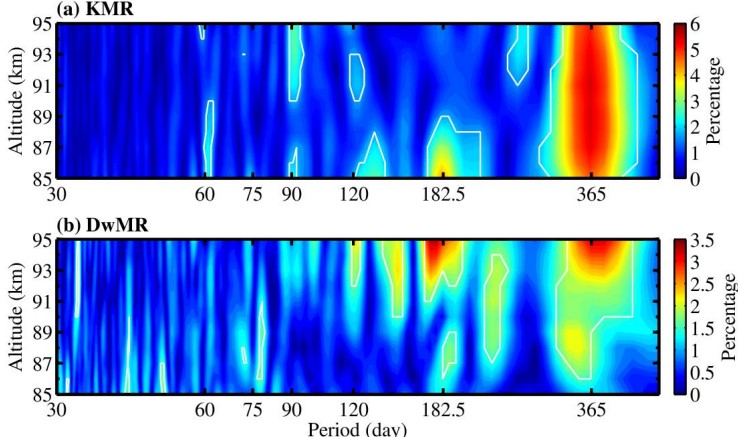

**Figure 8.** Same as Figure 4, but for the KMR and DwMR, daily mean densities at low latitudes.

## 4. Composite analysis for the global mesopause density

In the results described above, we presented the year-to-year variability in the climatology of the global mesopause density. To better appreciate the latitudinal changes of the seasonal variations in the global mesopause density, we show a composite analysis for the nine meteor radar measurements in Figure 9. For this composite analysis, we first combine the nine meteor radar densities into a single year and then use a 30-day running average to obtain the seasonal variations in the global mesopause density. As shown in Figure 9, several distinct features are present in the climatology of the global mesopause density.

It is clear that the seasonal variations in the mesopause densities exhibit latitudinal



differences. The seasonal variations in the mesopause densities obtained from the SMR and TMR at northern high latitudes and the MMR at higher northern mid-latitudes are similar; they display a primary maximum after the spring equinox and a minimum during the summer. In the northern mid-latitudes, the mesopause

densities from the BMR, McMR and WMR exhibit similar seasonal variations with a strong maximum near the spring equinox, a weak maximum before the winter solstice, and a minimum during the summer. As shown in Figure 9, the most noticeable feature is that the temporal evolution of the maximum mesopause density shifts as the latitude changes. For instance, the phase of the maximum shifts from spring (May) to winter

(January) across the spring equinox from the high latitudes to the low latitudes in the Northern Hemisphere. Referring to the recent studies by Jia et al. (2018) and Ma et al. (2018), a similar feature was also present in the zonal mean winds simultaneously observed by the MMR, BMR, McMR and WMR at northern mid-latitudes; they reported that the zonal winds above 85 km generally exhibit an annual variation with

a maximum during the summer (eastward), and they further demonstrated that the wind shifts (i.e., the zero zonal wind) near the spring equinox. In addition, based on their results, we also find that the phase of the maximum in the zonal wind also shifts as the latitude decreases; meanwhile, the time at which the zonal wind shifts also demonstrates a transition across the spring equinox from the MMR to the WMR,

which is similar to the observed mesopause densities shown in Figure 9.



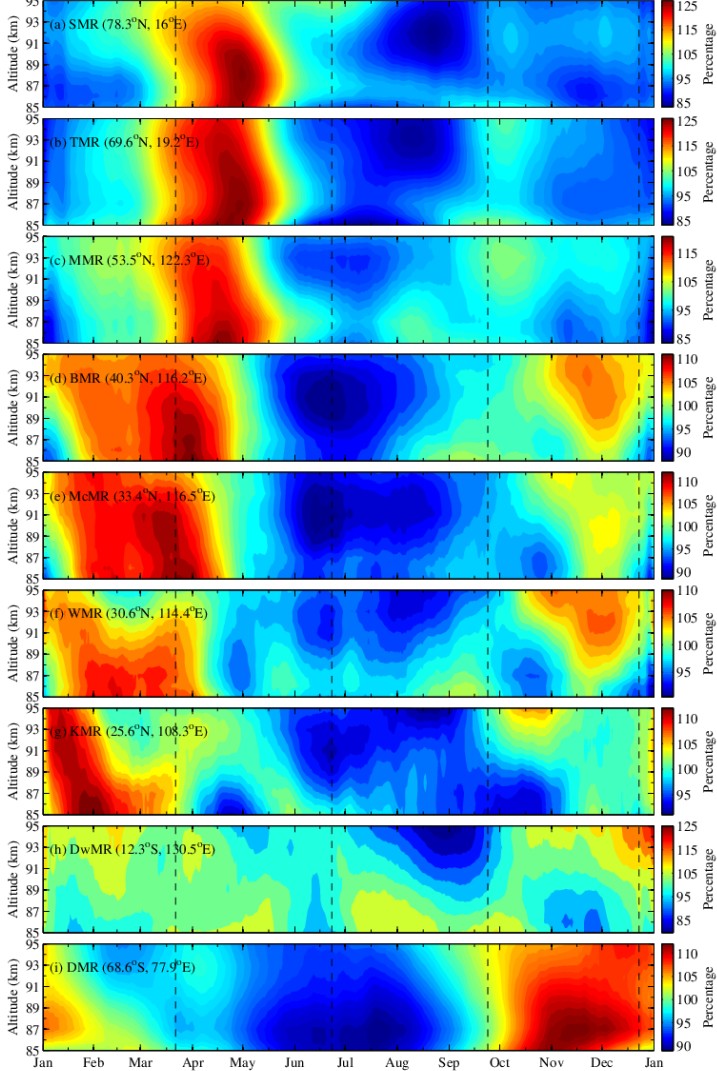

**Figure 9.** Contours of the composite 30-day running mean values of the mesopause densities in the composite year from the north pole to the south pole observed by the (a) SMR, (b) TMR, (c) MMR, (d) BMR, (e) McMR, (f) WMR, (g) KMR, (h) DwMR, and (i) DMR. The dashed lines indicate the spring and autumn equinoxes and the summer and winter solstices. The colour bars indicate the percentage variation in the 30-day running mean density relative to the mean density from the total observational time period.





It is also worth noting that the minima of the global mesopause densities appear during June, July and August. The minima of the northern polar mesopause densities obtained from the SMR and TMR occur during the Northern Hemisphere summer, while the DMR densities also show minima during the Southern Hemisphere winter.

The mesopause densities over the northern mid-latitudes obtained from the MMR, BMR, McMR and WMR all appear during the Northern Hemisphere summer. Because no measurements of the mesopause density over the southern mid-latitudes are presented in this study, we cannot provide a comparison for the interhemispheric mid-latitudes. With regard to the low latitudes, the mesopause densities obtained from

the KMR clearly show a minimum during the Northern Hemisphere summer above 87 km. In contrast, the DwMR densities in the southern low latitudes show a clear minimum during August and September, which is not during the expected Southern Hemisphere summer. These results reveal a seasonal asymmetry in the mesopause density in both hemispheres. During the Northern Hemisphere summer (the perihelion

is on July 4), the distance between the Sun and the Earth is 3.3% longer than that during the Northern Hemisphere winter (the aphelion is on January 3); therefore, the longer distance between the Sun and the Earth during the Northern Hemisphere summer leads to a reduction of 6.7% in the total solar radiation absorbed by the Earth, causing the Earth's atmosphere to shrink. This may explain why the global mesopause

densities show a minimum during the Northern Hemisphere summer.

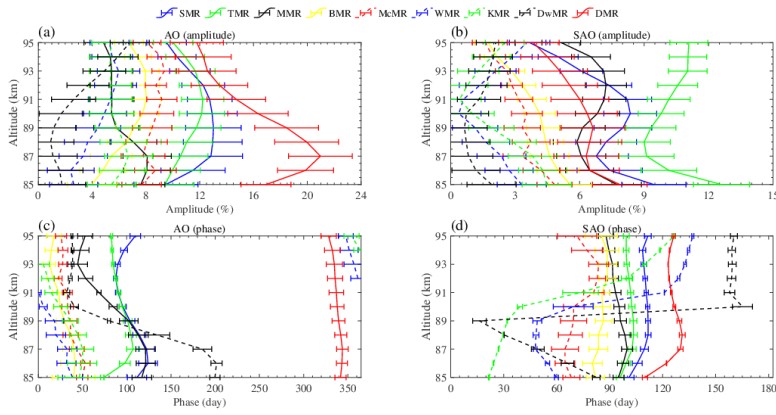





**Figure 10.** Amplitudes (top) and phases (bottom) of the AO and SAO observed by the nine meteor radars. The amplitude values indicate the percentage of the density relative to the mean density from the total observational time period.

Figure 10 shows the harmonic fitting results for the composite global mesopause density (shown in Figure 9). As shown in Figure 10a, it is clear that the AO displays large amplitudes exceeding 10% at high latitudes (DMR, SMR and TMR); the maxima of the AO amplitudes observed by the DMR, SMR and TMR reach 21%, 13% and 12%, respectively. Moreover, the amplitudes of the AO at southern high latitudes (DMR) are much larger than those at northern high latitudes (SMR and TMR). In the mid-latitudes (MMR, MBR, McMR and WMR), the AO amplitudes observed by the McMR are stronger than those observed by the other three stations, especially the MMR and BMR situated in the higher mid-latitudes. At low latitudes, the AO observed by the KMR is stronger than that observed by the DwMR at lower latitudes in the Southern Hemisphere as well as that observed by the WMR at higher latitudes.

Similarly, Figure 10b shows the SAO amplitudes observed by the nine meteor radars; the SAO is much weaker than the AO, as shown in Figure 10a. It is clear that the SAO is strongest at the TMR and that the amplitudes are comparable to those of the AO with a mean of approximately 10%. The SAOs in the northern high latitudes (SMR and TMR) are stronger than those in the southern high latitudes (DMR). In the mid-latitudes (MMR, BMR, McMR and WMR), the amplitudes of the SAOs decrease as the latitude decreases and roughly increase with decreasing altitude. The SAOs are much weaker in the low latitudes (KMR and DwMR), which is different from the temperature and horizontal wind in the low-latitude mesopause. The SAO is clearly the dominant seasonal variation in both the horizontal wind (Li et al., 2012) and the temperature (Xu et al., 2007) in the mesosphere at low latitudes. This might be because the seasonal variations in the mesopause density are influenced by the atmospheric dynamics as well as atmospheric equilibrium; however, this relationship is too complicated to understand at the moment.

Figures 10c and 10d show the phases of the AO and SAO, respectively, observed by



the nine meteor radars. The phases of the AO show an approximately decreasing trend as the latitude decreases and a downward progression as the altitude increases. In addition, the phases of the SAO clearly show a decreasing trend from the high latitudes (SMR) to the low latitudes (KMR); this can also explain the shift in the

temporal evolution of the mesopause density maxima as the latitude changes. The times at which the density maxima occur (Figure 9) are consistent with the phases of the SAO shown in Figure 10d. In addition, the phases of the SAO observed by the WMR, KMR and DwMR show a phase shift as the altitude increases; this is also reflected in Figure 9. Placke et al. (2011) and Jia et al. (2018) calculated the gravity

wave momentum fluxes in the mesosphere and lower thermosphere using the meteor radars at Collm, Germany (51.3 °N, 13.0 °E), Mohe and Beijing; they reported that the gravity wave variations exhibit an SAO at an altitude of approximately 90 km with a maximum during the summer and a secondary, weaker maximum during the winter as well as two minima around the equinoxes. Furthermore, Dowdy et al. (2001)

suggested that radiative effects are stronger in the Southern Hemisphere and that gravity wave driving effects are more important in the Northern Hemisphere. These results may explain why the SAOs are more obvious at the SMR and TMR at high latitudes and at the MMR and BMR at higher mid-latitudes as well as why the SAO at northern high latitudes is stronger than that at southern high latitudes.





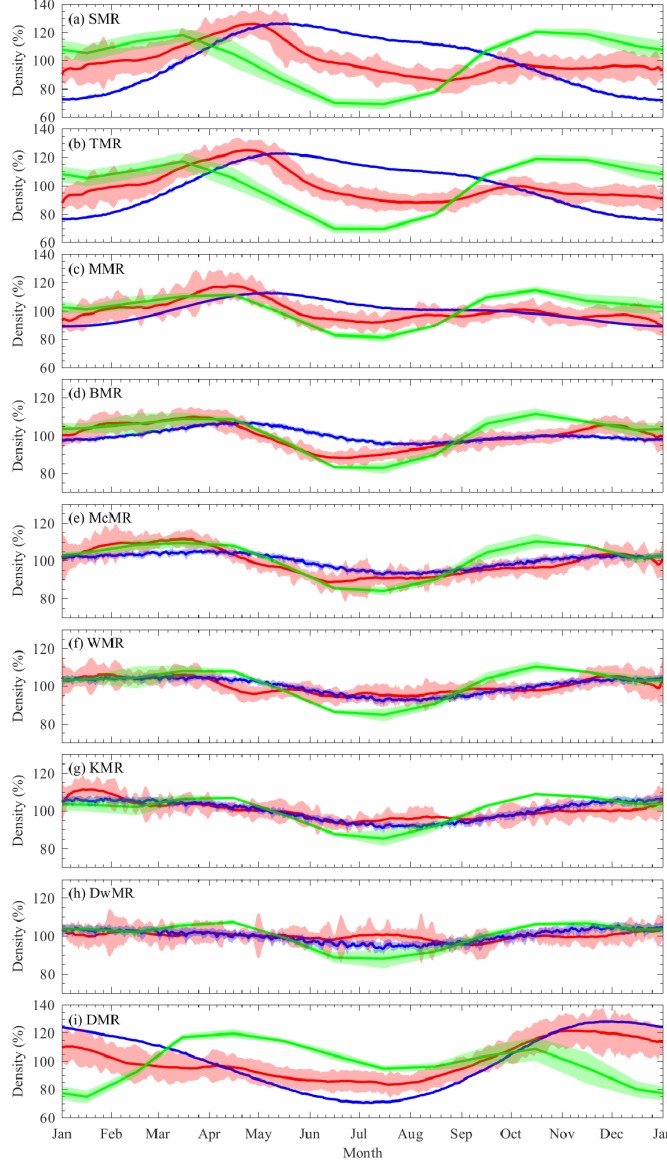

**Figure 11.** Comparisons of the mesopause densities at 90 km in the composite year among the meteor radars (red solid lines), the Mass Spectrometer and Incoherent Scatter (MSIS) model (blue solid lines) and the Whole Atmosphere Community Climate Model (WACCM) (green solid lines). The shaded areas represent the 30-day running averages and standard deviations of the composite density.



Figure 11 shows a comparison of the climatology of the mesopause density at 90 km in the composite year among the meteor radars in addition to the mesopause densities calculated simultaneously by the U.S. Naval Research Laboratory Mass Spectrometer and Incoherent Scatter (NRLMSISE-00) model (Picone et al., 2002) and Whole

Atmosphere Community Climate Model version 4 (WACCM4). The WACCM is an atmospheric component of the Community Earth System Model (CESM) version 1.0.4 developed by the National Center for Atmospheric Research; the key features are described in detail in Marsh et al. (2013). In addition, the WACCM is a superset of the Community Atmospheric Model version 4 with 66 vertical hybrid levels from the

surface to the lower thermosphere (~145 km); the vertical spacing increases with the altitude from ~1.1 km in the troposphere to 1.1–1.8 km in the lower stratosphere and 3.5 km above ~65 km. The horizontal resolution for the WACCM4 used here is 1.9° latitude by 2.5° longitude.

The comparisons shown in Figure 11 reveal evident differences between the

observations and models. The MSIS densities show a dominant AO, the amplitude of which decreases as the latitude decreases. In the southern high latitudes, the MSIS densities generally exhibit an annual variation similar to those displayed by the DMR observations with a maximum during November and December and a minimum during July, but the AO shows a larger variation than do the DMR observations. In the

Northern Hemisphere from the SMR to the McMR, the difference between the meteor radar observations and the MSIS model is obvious because the SAOs in the meteor radar observations are strong at these latitudes, while the SAO amplitude is much weaker in the MSIS model. At lower latitudes, the MSIS captures only the annual variations in the WMR, KMR and DwMR observations but fails to reproduce the

other seasonal and intraseasonal variations. The WACCM densities show mainly annual and semiannual variations but almost fail to capture the seasonal variations in the mesopause density. However, it is worth noting that the WACCM densities show a minimum during June, July and August; this feature is similar to the meteor radar observations. The comparison between the observations and models demonstrates

obvious inconsistencies, which indicate some limitations of the current models, such





as the MSIS model and WACCM, regarding the seasonal behaviour of the mesopause density.

The MSIS model is an empirical atmospheric model based on observations acquired over a decade ago; in particular, mesospheric density data were quite scarce at that time, which is the likely reason that the MSIS model exhibits obvious differences from the meteor radar observations. Moreover, the WACCM cannot directly provide atmospheric density estimates. Hence, in this study, we calculate the WACCM density at 90 km using the WACCM-simulated temperature and the geographic height corresponding to the pressure level. Previous studies indicated that WACCM-simulated temperatures are generally higher than lidar observations, but the WACCM temperatures can reproduce the major features of the climatology of the mesopause temperatures (see, e.g., Li et al., 2018). The accuracy of the pressure level (i.e., geographic height) is quite difficult to estimate because of the lack of corresponding observations. This study constitutes the first time we have compared the mesopause density simulated by the WACCM with meteor radar observations; hence, the remarkable differences in the seasonal variations between them are difficult to understand at the moment and are beyond the scope of this study.

## 5. Summary

Mesopause densities determined with data from a global distribution of meteor radars are used to investigate the climatology of the global mesopause density. The multiyear observations of the mesopause density involved nine meteor radars, namely, the Davis Station (68.6 °S, 77.9 °E), Svalbard (78.3 °N, 16 °E) and Tromsø (69.6 °N, 19.2 °E) meteor radars located at high latitudes, the Mohe (53.5 °N, 122.3 °E), Beijing (40.3 °N, 116.2 °E), Mengcheng (33.4 °N, 116.6 °E) and Wuhan (30.5 °N, 114.6 °E) meteor radars located in the mid-latitudes, and the Kunming (25.6 °N, 103.8 °E) and Darwin (12.4 °S, 130.8 °E) meteor radars located at low latitudes. The mesopause densities estimated from these nine meteor radars exhibit different seasonal and latitudinal variations. The main points of the latitudinal and seasonal variations in the mesopause density are summarized as follows:



1. In the southern high latitudes, the AO observed by the DMR dominates the seasonal variations with a maximum during the late spring and a minimum during the early winter. In the northern hemisphere from high to low latitudes (from the SMR to the KMR), the AOs dominate the seasonal variations in the mesopause densities, and the amplitudes decrease equatorward. In addition to AOs, SAOs are also evident in the Northern Hemisphere, especially at high latitudes, and their largest amplitude, which is detected at the TMR, is comparable to the AO amplitudes. Near the equator, the mesopause densities observed by the DwMR show an AO and relatively weak intraseasonal oscillations with a periodicity of 30-90 days, which are related to the MJO in the tropical troposphere.

2. Interhemispheric observations indicate that the mesopause densities over the southern and northern polar regions show a clear seasonal asymmetry. The maxima of the yearly variations in the mesopause density exhibit a clear temporal variation across the spring equinox as the latitude decreases; these latitudinal variation characteristics may be related to the latitudinal variation in the global circulation of the mesosphere influenced by gravity wave forcing. In addition, the minima of the global mesopause densities basically appear during June, July and August, rather than a conditioned to think that the seasonal symmetric in interhemispheric temperature and wind. A possible explanation for this phenomenon is that the longer distance between the Sun and the Earth during the Northern Hemisphere summer leads to a reduction in the total solar radiation absorbed by the Earth that then causes the Earth's atmosphere to shrink. However, the actual mechanism cannot be comprehensively proven at the moment and thus remains an open question. Future observations and modelling are needed to more completely characterize and explain these phenomena.

3. Comparisons of the climatology of the mesopause density at 90 km among the observations from meteor radars are provided in addition to the mesopause densities calculated simultaneously by the MSIS model and WACCM. The MSIS model roughly captures the prevailing annual variation in the mesopause density at southern high latitudes and northern low latitudes. The WACCM densities show both annual



and semiannual variations but almost fail to capture the seasonal variations in the mesopause density. The comparison results show the above inconsistencies between the observations and models, thereby indicating some limitations of the current models, such as the MSIS model and WACCM, regarding the seasonal behaviour of

the mesopause density.

In this study, we have reported global observations of the climatology of the mesopause density for the first time. Knowledge of the atmospheric density is essential for understanding the relevant physical processes in the mesopause region as well as for providing a usual reference for lidars (e.g., Dou et al., 2009) or an input

parameter for the airglow phenomenon (Reid et al., 2017; Takahashi et al., 2002). However, accurately predicting the changes in the neutral atmospheric density over time is crucial for determining the atmospheric drag on low-Earth-orbit satellites and directly governs the orbit cycles of satellites; moreover, safe launches and precise spacecraft landings also require accurate knowledge of the neutral atmospheric

density. Despite the differences between the observations and model simulations, the mesopause densities derived from meteor radar observations still have great potential and practical applications because the global distribution of meteor radar instruments and their associated long-term and continuous datasets provide a wide range of aerospace applications and the potential to improve widely used empirical models.

*Data availability*. The Aura/MLS data are available from http://disc.sci.gsfc.nasa.gov/Aura/data-holdings/MLS. The Davis meteor radar data are available from the Australian Antarctic Data Centre at data.aad.gov.au. The Svalbard and Tromsø meteor radar data are available upon request from Chris Hall at Tromsø Geophysical Observatory (chris.hall@uit.no). The Mohe, Beijing and Wuhan meteor

radar data are available from http://data.meridianproject.ac.cn/. The Mengcheng and Kunming meteor radar data are available upon request from Wen Yi (yiwen@ustc.edu.cn).

*Author contributions*. WY designed the study, performed data analysis, prepared the figures and wrote the manuscript. XX initiated the study, and contributed to





supervision and interpretation. IMR contributed to supervision and interpretation, and help for writing-original editing. DJM provided the Davis meteor radar data. CMH and MT provided the Svalbard and Tromsø meteor radar data. NB and LG provided the Mohe, Beijing and Wuhan meteor radar data. RV provided the Darwin meteor

radar data. CJ provided the Kunming meteor radar data. WJ are responsible for the WACCM model. CT and DX contributed to interpretation. All authors contributed to discussion and interpretation.

*Competing interests.* The authors declare that they have no conflict of interest.

*Acknowledgments.* This work is supported by the National Natural Science

Foundation of China (41774158, 41474129, and 41674150), the Chinese Meridian Project, the Youth Innovation Promotion Association of the Chinese Academy of Sciences (2011324), and the China Scholarship Council. We also acknowledge support provided by the University of Adelaide and ATRAD Pty Ltd, and the provision of Davis meteor radar data by the Australian Antarctic Division, the

provision of Nippon/Norway Svalbard and Tromsø meteor radar data by the National Institute of Polar Research and UiT-The Arctic University of Norway, the provision of Mohe, Beijing and Wuhan meteor radar data by the Chinese Meridian Project and STERN (the Solar: Terrestrial Environment Research Network), the provision of Kunming meteor radar data by the China Research Institute of Radiowave

Propagation (CRIRP), the provision of Darwin meteor radar data by the University of Adelaide. Operation of the Davis meteor radar was supported under AAS project 2529, 2668, and 4025.

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
