# Peer review of "Climatology of the mesopause relative density using a global distribution of meteor radars"

_Atmospheric Chemistry and Physics, 2018_

## Referee Comment (RC1) · Anonymous Referee #1 · 31 Dec 2018

The work is a comprehensive analysis of atmospheric density based on several meteor radar measurements at various latitudes. It is the first study of this kind that provides a near-global view of mesopause density variations based on ground-based instruments. The analysis is thorough and the figures and presentations are very clear. The discrepancies with MSIS and WACCM are especially interesting and should motivate further research to understand them. I recommend publication of this work after the following comments are addressed:

Major:

One factor that could case large seasonal variations is different heights the density is inferred from. Although the meteor radar data used are all from 85-95 km range, there is clearly seasonal variations of the vertical distributions of detected meteors, as

shown in Figure 2(a). Thus the derived densities are weighted differently in vertical, which may introduce a 'false' seasonal variation. Please address this issue carefully so the comparison with MSIS and WACCM can be more appropriate.

Minor:

P2L23: This sentence is confusing. How can a 'maxima of a yearly variation' still has 'temporal variation'? 'as the latitude decreases' seems to suggest the variation is with latitude, not time. Please clarify.

P2L28 (and other places). The 30-60 days oscillation found is similar to that of MJO, but there is no evidence in this analysis that it is actually due to MJO. It is only speculation so claiming that it is "related to' MJO is too strong a claim.

---

## Referee Comment (RC2) · Anonymous Referee #2 · 28 Jan 2019

**GENERAL COMMENTS**

The mesosphere and lower thermosphere (MLT) is the boundary between the middle atmosphere and the upper atmosphere. Physical processes in the MLT determine the fluxes of waves and tides that propagate into the thermosphere and so act to influence the coupling of these atmospheric regions. There is thus a need for measurements able to characterise the properties of the MLT. Measurements of winds and temperatures in the MLT have been made for many years by radars, lidars and satellites. However, it is very difficult to measure densities at these heights and such measurements are particularly valuable.

This paper presents observations of MLT densities made by combining radar measurements of ambipolar diffusion coefficient with satellite measurements of temperature. A

total of nine radars are used, giving coverage over all latitudinal regimes apart from mid-latitudes in the SH.

The key results of the paper are determinations of the seasonal cycle in MLT density over the different radars, which also reveals interesting inter-hemispheric differences and suggestions of MLT perturbations associated with the MJO. Comparisons are made with models which highlight deficiencies in the models.

The paper is exceptionally well written and was a pleasure to read. The analysis is persuasive and is presented in a logical manner. The figures are appropriate, easy to understand and nicely produced. The references are adequate and up to date. The abstract is clear and accessible.

Overall, this is scientific work of a high standard which presents interesting and significant results. It merits publication in ACP subject to a few minor clarifications as detailed below.

SPECIFIC COMMENTS

1. The authors determine ambipolar diffusion coefficient at each height from the radar data. Is there any sorting of data by elevation angle? Meteors recorded at low elevation angle will be at long ranges and so even a small error in elevation angle may thus correspond to a significant error in height. Conversely, errors in elevation angle will produce smaller errors in height near the zenith. The authors should explain if they used any sorting and comment on this possibility.

2. The authors use MLS temperatures in combination with the ambipolar diffusion to estimate density. However, at these heights the vertical resolution of MLS is poor, e.g., at z = 81 km the vertical resolution is $\sim$ 14 km. Given that the atmosphere at these height can have sharp temperature gradients associated with the mesospause, how does this impact the analysis? Is this not a major source of uncertainty in the determination of density given that the actual atmospheric temperature at a particular

height may be rather different from the one derived from MLS measurements of low resolution?

TECHNICAL CORRECTIONS

1. Please check all references are present – e.g., Dowdy et al. (2001) is mentioned on p3 but missing from the references.

2. The manuscript refers to "densities", but the measurements are actually "relative densities". This should be corrected throughout to avoid confusion.

---

## Author Comment (AC1) · 9 Mar 2019

We thank the **reviewer#1** for the useful suggestions to improve the paper. These comments are all valuable and very helpful for revising and improving our manuscript, as well as the important guiding significance to our researches. These changes in the revise manuscript have been **marked red** in the track changes version manuscript, as well as the point to point responses have listed as following:

**General comments:**

The work is a comprehensive analysis of atmospheric density based on several meteor radar measurements at various latitudes. It is the first study of this kind that provides a near-global view of mesopause density variations based on ground-based instruments. The analysis is thorough and the figures and presentations are very clear. The discrepancies with MSIS and WACCM are especially interesting and should motivate further research to understand them.

**Response:**

Thank you for your great comments. This will encourage us to improve our manuscript, as well as the important guiding significance to our researches.

**Major comment:**

One factor that could cause large seasonal variations is different heights the density is inferred from. Although the meteor radar data used are all from 85-95 km range, there is clearly seasonal variations of the vertical distributions of detected meteors, as shown in Figure 2(a). Thus the derived densities are weighted differently in vertical, which may introduce a 'false' seasonal variation. Please address this issue carefully so the comparison with MSIS and WACCM can be more appropriate.

**Response:**

The number of meteor echoes in vertical might influence the calculation of the ambipolar diffusion coefficients, but the influence is very small and should not cause a seasonal variation in vertical. First, we actually did compare the estimation of Da using different meteor detection in this study. As shown in Figure 2 in the manuscript, we have shown the estimation of Da by using two meteor detections with different

number observed by the Davis 33 MHz and 55 MHz meteor radars. Although the number of 55 MHz meteor echoes is much lower than the 33 MHz meteor radar, it still provides a precise estimation of Da. In the manuscript, this indirectly indicates that the seasonal variations of meteor detections should not introduce a 'false' seasonal variation.

In addition, in this response, in order to estimate the bias in daily mean Da caused by the meteor with different numbers, we also estimated the Da using different meteor detections observed by the Mohe meteor radar. As shown in Figure R1a, the mean values of Da at 90 km estimated by using all, 1/2 and 1/4 of meteor detections are quite consistent. The linear correlation coefficients between all and half, and 1/4 meteors are 0.995 and 0.985, respectively. These results in above also indicate that the influence of meteor number is small and should not cause a seasonal variation.

[Figure]

**Figure R1. (a)** Daily mean ambipolar diffusion coefficients at 90 km in 2013 estimated from the Mohe meteor radar. The red solid, green dashed and blue dotted lines indicate the mean values calculated by using all, 1/2 and 1/4 of meteor echoes in 1 km bin from 89.5 to 90.5 km, respectively. **(b)** Comparison of daily mean ambipolar diffusion coefficients at 90 km estimated from all and 1/2 of meteor echoes. **(c)** same as (b), but for 1/4 of meteor echoes.

**Minor comments:**

P2 L23: This sentence is confusing. How can a 'maxima of a yearly variation' still has 'temporal variation'? 'as the latitude decreases' seems to suggest the variation is with latitude, not time. Please clarify.

**Response:** Thank you for pointing this out. We have changed in the revised manuscript.

**The sentence is changed as:** The maxima of the yearly variations in mesopause densities have a clear latitudinal variation, across forward the spring equinox, as the latitude decrease; these latitude variation characteristic may relate to the latitudes changes of the gravity wave forcing.

P2 L28 (and other places). The 30-60 days oscillation found is similar to that of MJO, but there is no evidence in this analysis that it is actually due to MJO. It is only speculation so claiming that it is "related to' MJO is too strong a claim.

**Response:** Thank you for your suggestion, we have made a change in the abstract.

**The sentence is changed as:** In addition to the AO, the mesopause densities over low latitudes also clearly show an intraseasonal variation with a periodicity of 30-60 days.

In addition to the intraseasonal variations in Darwin density, as shown in Figure R2 and R3, we also find obvious intraseasonal variations in zonal mean wind observed by the Darwin meteor radar and the periodogram in density is consistent with zonal mean wind. However, this is beyond the scope of this paper, and a more detailed discussion of this intercomparison will be described in a forthcoming paper.

[Figure]

**Figure R2.** Residual of zonal mean wind observed from Darwin meteor radar.

[Figure]

**Figure R3.** (a) Residual of zonal mean wind at 80 km observed by the Darwin meteor radar. (b) Wavelet power spectrum of the zonal mean wind. The black solid contours denote the regions of the wavelet spectrum above 95% confidence level. (c) Lomb-Scargle periodogram for the zonal mean wind. The blue dashed line represents the 95% significance level of the Lomb-Scargle periodogram.

---

## Author Comment (AC2) · 9 Mar 2019

We thank the **reviewer#2** for the useful suggestions to improve the paper. These comments are all valuable and very helpful for revising and improving our manuscript, as well as the important guiding significance to our researches. These changes in the revise manuscript have been **marked red** in the track changes version manuscript, as well as the point to point responses have listed as following:

**General comments:**

The mesosphere and lower thermosphere (MLT) is the boundary between the middle atmosphere and the upper atmosphere. Physical processes in the MLT determine the fluxes of waves and tides that propagate into the thermosphere and so act to influence the coupling of these atmospheric regions. There is thus a need for measurements able to characterise the properties of the MLT. Measurements of winds and temperatures in the MLT have been made for many years by radars, lidars and satellites. However, it is very difficult to measure densities at these heights and such measurements are particularly valuable.

This paper presents observations of MLT densities made by combining radar measurements of ambipolar diffusion coefficient with satellite measurements of temperature. A total of nine radars are used, giving coverage over all latitudinal regimes apart from mid-latitudes in the SH. The key results of the paper are determinations of the seasonal cycle in MLT density over the different radars, which also reveals interesting inter-hemispheric differences and suggestions of MLT perturbations associated with the MJO. Comparisons are made with models which highlight deficiencies in the models.

The paper is exceptionally well written and was a pleasure to read. The analysis is persuasive and is presented in a logical manner. The figures are appropriate, easy to understand and nicely produced. The references are adequate and up to date. The abstract is clear and accessible. Overall, this is scientific work of a high standard which presents interesting and significant results.

**Response:** Thank you for your great comments. This will encourage us to improve our manuscript, as well as the important guiding significance to our researches.

**Minor comment:**

1. The authors determine ambipolar diffusion coefficient at each height from the radar data. Is there any sorting of data by elevation angle? Meteors recorded at low elevation angle will be at long ranges and so even a small error in elevation angle may thus correspond to a significant error in height. Conversely, errors in elevation angle will produce smaller errors in height near the zenith. The authors should explain if they used any sorting and comment on this possibility.

**Response:** Thank you for pointing this out. We apologize that we forgot to refer to data processing in this study. In the data processing, we actually did the zenith selection for the meteor echoes. The data processing is followed by Yi et al. (2017, 2018a). In order to avoid the possibility of excessive error in the height estimates of individual meteors, trail detections for this study were restricted to zenith angles of less than 60 °.

**We have added this in the revised manuscript.** In order to avoid the possibility of excessive error in the height estimates of individual meteors, trail detections for this study were restricted to zenith angles of less than 60 °.

2. The authors use MLS temperatures in combination with the ambipolar diffusion to estimate density. However, at these heights the vertical resolution of MLS is poor, e.g., at z = 81 km the vertical resolution is _ 14 km. Given that the atmosphere at these height can have sharp temperature gradients associated with the mesopause, how does this impact the analysis? Is this not a major source of uncertainty in the determination of density given that the actual atmospheric temperature at a particular height may be rather different from the one derived from MLS measurements of low resolution?

**Response:** Thank you for pointing this out. First, we should point out that we chose the MLS temperature measurements because the Aura/MLS has very good temporal continuity in polar region. Schwartz et al. (2008) suggested that the vertical resolution of geopotential height (GPH) is ~13 km at 0.001 hPa, however, the precision is ±100 m, and the observed bias is -450 m. In addition, in the response, we also compared the geopotential heights from the MLS and SABER measurements. Figure R1 shows the

daily geometric heights at pressure level 0.00464 (red), 0.00215 (green), 0.001 (blue) and 0.000464 (black) hPa calculated from the MLS geopotential heights compare to the geometric heights estimated from SABER pressure measurements over Beijing. The Geometric heights, $z$ for Aura MLS observations were computed from geopotential heights, $z_g$ via the equation $z = z_g R_e(\emptyset)[R_e(\emptyset) - z_g]^{-1}$, where $R_e(\emptyset)$is the radius of Earth at latitude $\phi$, based on the WGS84 ellipsoid [*Decker*, 1986]. As shown in Figure R1, the seasonal variations of the geometric height at each pressure level from the MLS and SABER measurements are consistent. The biases between two geometric heights are less than 1km at pressure level 0.00464, 0.00215, 0.001 hPa and are about 2 km at pressure level 0.000464 hPa. So these results confirm that the geopotential height is precise enough and should not introduce major uncertainty in temperature interpolation.

[Figure]

**Figure R1.** Comparison of the geometric heights at pressure level 0.00464 (red), 0.00215 (green), 0.001 (blue) and 0.000464 (black) hPa from Aura/MLS (plus sign) and TIMED/SABER (circle sign) measurements from 2012 to 2016.

TECHNICAL CORRECTIONS

1. Please check all references are present – e.g., Dowdy et al. (2001) is mentioned on p3 but missing from the references.

**Response:** Thank you for pointing this out. We have added the missing reference (i.e., Dowdy et al. 2001) in the revised manuscript.

Reference: Dowdy, A., Vincent, R., Igarashi, K., Murayama, Y., and Murphy, D.: A comparison of mean winds and gravity wave activity in the northern and southern polar MLT. Geophys Res Lett, 28, 1475-1478, 2001.

2. The manuscript refers to "densities", but the measurements are actually "relative densities". This should be corrected throughout to avoid confusion.

**Response:** Thank you for pointing this out, we have corrected throughout in the revised manuscript.

References in this response:

Decker, B.: World Geodetic System 1984, Def. Mapp. Agency Aerosp. Cent., St. Louis AFS, Mo, 1986.

Schwartz, M., Lambert, A., Manney, G., Read, W., and Livesey, N.: Validation of the Aura Microwave Limb Sounder temperature and geopotential height measurements, J. Geophys. Res., 113, D15S11, doi:10.1029/2007JD008783, 2008.

Yi W., Reid, I., Xue, X., Younger, J., Murphy, D., Chen, T., and Dou, X.:, Response of neutral mesospheric density to geomagnetic forcing, Geophys. Res. Lett., doi: 10.1002/2017GL074813, 2017.

Yi, W., Reid, I., Xue, X., Murphy, D., Hall, C., Tsutsumi, M., Ning, B., Li, G., Younger, J., Chen, T., and Dou, X.: High- and middle-latitude neutral mesospheric density response to geomagnetic storms. Geophysical Research Letters, 45, 436–444. https://doi.org/10.1002/2017GL076282, 2018a.

---

## Author Comment (AC4) · 9 Mar 2019

This is for manuscript with track change turned off, i.e. clean version. Please download supplement pdf file.

Please also note the supplement to this comment:
https://www.atmos-chem-phys-discuss.net/acp-2018-1040/acp-2018-1040-AC4-supplement.pdf

---

## Author Response (AR2)

**Response to the Editor's comments**

**Comments**

Reviewer Comment: P2 L28 (and other places). The 30-60 days oscillation found is
similar to that of MJO, but there is no evidence in this analysis that it is actually due
to MJO. It is only speculation so claiming that it is "related to' MJO is too strong a
claim.

Author Response: Thank you for your suggestion, we have made a change in the
abstract. The sentence is changed as: In addition to the AO, the mesopause densities
over low latitudes also clearly show an intraseasonal variation with a periodicity of
30-60 days.

Editor comment: the assertion that the intraseasonal density variations are related to
the MJO is repeated on page 24, line 10 and has not yet been corrected.
**Author response: We have corrected in the revised manuscript.**

Reviewer Comment: The authors determine ambipolar diffusion coefficient at each
height from the radar data. Is there any sorting of data by elevation angle? Meteors
recorded at low elevation angle will be at long ranges and so even a small error in
elevation angle may thus correspond to a significant error in height. Conversely,
errors in elevation angle will produce smaller errors in height near the zenith. The
authors should explain if they used any sorting and comment on this possibility.

Author Response: Thank you for pointing this out. We apologize that we forgot to
refer to data processing in this study. In the data processing, we actually did the zenith
selection for the meteor echoes. The data processing is followed by Yi et al. (2017,
2018a). In order to avoid the possibility of excessive error in the height estimates of
individual meteors, trail detections for this study were restricted to zenith angles of
less than 60°. We have added this in the revised manuscript: In order to avoid the
possibility of excessive error in the height estimates of individual meteors, trail
detections for this study were restricted to zenith angles of less than 60°.

Editor comment: Please provide some estimate of the radars' elevation angle
measurement error, and how this will impact the height determination in the worst
case scenario of a meteor trail recorded at a zenith angle of 60 degrees.
**Author response: In this study, all meteor radars transmit a 3.6 km long, 4 bit
complimentary coded pulse with a pulse repetition frequency (PRF) of 430 Hz, so
the meteor radar range sampling resolution is 1.8 km (Holdsworth et al., 2008).
With the criterion of zenith<60°, the meteor height estimate uncertainty (range
sampling resolution×cos(zenith)) should be less than ±1 km.**
**We have added the error estimation on page 6 line 14 in the revised manuscript.**

Reviewer Comment: The authors use MLS temperatures in combination with the ambipolar diffusion to estimate density. However, at these heights the vertical resolution of MLS is poor, e.g., at z = 81 km the vertical resolution is 14 km. Given that the atmosphere at these height can have sharp temperature gradients associated with the mesopause, how does this impact the analysis? Is this not a major source of uncertainty in the determination of density given that the actual atmospheric temperature at a particular height may be rather different from the one derived from MLS measurements of low resolution?

Author Response: First, we should point out that we chose the MLS temperature measurements because...

Editor comment: I don't think the referee was implying that the MLS data are inaccurate, nor that the data coverage is poor in the polar regions. Rather that the MLS temperature measurements are an average value derived over 14 km of altitude at z=81km. As the referee correctly points out there can be large temperature gradients in the atmosphere at these altitudes. Some quantitative estimate of the uncertainties this may introduce in your relative density measurements - especially the difference between summer and winter - is required here.

**Author response: The vertical resolution near mesopause region (about 90 km) is ~3-4 km. This may introduce an uncertainty in the determination of density, because the interpolated temperature has a bias between the actual mesopause temperature. However, in the present study, we just examine the relative density, the bias between the MLS and SABER temperatures should not cause large difference. In this response, as shown in Figures R1a and R1b, we directly compare the mesopause relative MLS and SABER temperatures. These two monthly mean relative temperature show a very similar variation, and Figure R1c shows the relative temperature differences between them are less than 3%.**
**We have added the discussion of temperature difference on page 9 line 10 in the revised manuscript.**

[Figure]

**Figure R1. Variations in the monthly mean relative the (a) MLS and (b) SABER temperatures from 85 to 95 km over the Mohe meteor radar, (c) the relative difference between the MLS and SABER temperatures.**

Reviewer Comment: The manuscript refers to "densities", but the measurements are actually "relative densities". This should be corrected throughout to avoid confusion.

Author Response: Thank you for pointing this out, we have corrected throughout in the revised manuscript.

Editor comment: This has been corrected in five places (page 9, line 17; page 12, line 6) and in the captions to figures 3, 5 and 7. The rest of the manuscript continues to discuss "densities" and not the preferred "relative densities". As the referee correctly discusses, the measurements reported are relative densities (or density variations). At no point are absolute densities reported. The authors should clarify this important point throughout the manuscript (including in the paper title and the abstract).
**Author response: We have changed in the revised manuscript.**

**Track changes**

[revised manuscript text omitted]